# Cloning and Functional Verification of *CYP408A3* and *CYP6CS3* Related to Chlorpyrifos Resistance in the *Sogatella furcifera* (Horváth) (Hemiptera: Delphacidae)

**DOI:** 10.3390/biology10080795

**Published:** 2021-08-18

**Authors:** Yanwei Ruan, Xinxian Liu, Changwei Gong, Yuming Zhang, Litao Shen, Hasnain Ali, Yanyan Huang, Xuegui Wang

**Affiliations:** 1Biorational Pesticide Research Lab, Chengdu Campus, Sichuan Agricultural University, Chengdu 611130, China; 2018301047@stu.sicau.edu.cn (Y.R.); youguqiu@163.com (C.G.); 2019301118@stu.sicau.edu.cn (Y.Z.); shen-litao@163.com (L.S.); 2019601004@stu.sicau.edu.cn (H.A.); 2State Key Laboratory of Crop Gene Exploration and Utilization in Southwest China, Sichuan Agricultural University, Chengdu 611130, China; 2018211033@stu.sicau.edu.cn

**Keywords:** *Sogatella furcifera*, chlorpyrifos, cytochrome P450, RNA interference

## Abstract

**Simple Summary:**

As an important enzyme system in organisms, P450 multi-functional oxidase not only participates in the metabolism and synthesis of substances in organisms but can also maintain the normal physiological functions of organisms under stress. As one of the important rice pests, the harm caused by white-backed planthoppers has been increasing in recent years. Although the application of chemical pesticides as one of the prevention and control measures can slow down the harm of white-backed planthoppers, its resistance is also rising rapidly. Among them, the generation of metabolic resistance dominated by the P450 enzyme is more common. In this study, we measured the expression of ten P450 gene in vivo situations with a background of chlorpyrifos resistance in white-backed planthoppers. After selecting the two genes with the highest expression, the function of these two genes in the pesticide resistance process was verified by RNA interference and provided a theoretical basis for a follow-up study of the molecular mechanism of the P450 gene mediated by pesticide resistance formation.

**Abstract:**

The white-back planthopper (WBPH), *Sogatella furcifera*, mainly harms rice and occurs in most rice regions in China and Asia. With the use of chemical pesticides, *S. furcifera* has developed varying degrees of resistance to a variety of pesticides. In our study, a chlorpyrifos-resistant population (44.25-fold) was built through six generations of screening with a sublethal dose of chlorpyrifos (LD_50_) from a field population. The expression levels of ten selected resistance-related P450 genes were analyzed by RT-qPCR and found that *CYP408A3* and *CYP6CS3* were significantly more expressed in the third instar nymphs of the XY17-G5 and XY17-G6 populations, about 25-fold more than the Sus-Lab strain, respectively (*p* < 0.01). To elucidate their molecular function in the development of resistance towards chlorpyrifos, we cloned two P450 full lengths and predicted their tertiary protein structures. *CYP408A3* and *CYP6CS3* were also downregulated after injecting *dsCYP408A3*, *dsCYP6CS3*, or their mixture compared to the control group. Moreover, the mortality rates of the *dsCYP6CS3* (91.7%) and the mixture injection treatment (93.3%) treated by the LC_50_ concentration of chlorpyrifos were significantly higher than the blank control group (51.7%) and *dsCYP408A3* injection treatment (69.3%) at 72 h (*p* < 0.01). Meanwhile, the P450 enzyme activities in the dsRNA treatments were lower than that in the control (XY17-G6) (*p* < 0.01). Therefore, the P450 gene *CYP6CS3* may be one of the main genes in the development of chlorpyrifos resistance in *S. furcifera*.

## 1. Introduction

The white-backed planthopper (WBPH), *Sogatella furcifera* (Horváth) (Hemiptera: Delphacidae), is among the most notorious insect pests of the rice crop in Asia. It usually feeds on rice stalks with its stinging mouthparts and can carry and spread southern Rice Black Streaked dwarf virus (RBSDV), which can harm the normal physiological growth of rice plants [1,2,3,4,5,6,7]. Over the past few decades, the most effective measure against *S. furcifera* is the use of pesticides. Consequently, one of the major crises is drug resistance caused by the inappropriate use and abuse of pesticides, as *S. furcifera* appears to have an extraordinary ability to develop resistance to a variety of insecticides, including chlorpyrifos [8,9,10,11,12]. A field WBPH resistance for chlorpyrifos was reported in Guizhou, Sichuan Province and many other rice growing areas of China in the past 15 years [11,12,13]. As a traditional and highly effective organophosphorus insecticide, Chlorpyrifos continues to be used in the control of pests such as Lepidoptera, Diptera, and Hemiptera [14]. However, due to the complex environment of pesticide use and the improvement of insect adaptability, organophosphate pesticides such as chlorpyrifos are suffering a new risk crisis of being banned. The pesticide industry is ambivalent about such a move, and chlorpyrifos is still a commonly used insecticide to control agricultural pests [15,16]. However, due to its stable nature and low utilization efficiency after its application (in the effective period of time to the full effect), many countries and regions are restricting the use of chlorpyrifos in order to reduce the excessive use of chlorpyriphos and reduce the generation of resistance in target species [17,18]. In addition, rice has now reached 363.3 million hectares since it was planted on a large scale in China. There is a buffering period for the replacement of insecticides, and chlorpyrifos is still used in many rice-growing areas. Therefore, it is still necessary to explore the mechanism of chlorpyrifos resistance through basic research in order to provide a theoretical basis for the subsequent research and development of biological pesticides and the promotion of new technologies [19,20].

Under natural conditions, when pest are subjected to long-term chemical stress, the development of resistance is a natural phenomenon in which the most significant factor is the enhanced metabolic activity of the enzymes related to insecticide resistance, such as mixed-function oxidases (MFO), carboxylesterases (CarEs), glutathione *S*-transferases (GSTs) [21], and particularly, the cytochrome P450 monooxygenases (P450s), which plays an important role in mediating the stress resistance of pests, such as insecticide metabolism [22,23]. Many studies have been performed relating an elevated P450 activity with an observed pesticide resistance in rice planthoppers. Zhang et al. [24] found that the four P450 genes played a major role in the metabolic resistance of *Nilaparvata lugens* (Stal) (Homoptera: Delphacidae) for about 25 generations, screened with the sublethal dosage of imidacloprid. Another study suggested that the overexpression of P450s could contribute to insecticide resistance in the field *Laodelphax striatellus* (Fallén) populations of Jiangsu Province, especially for the multiple types of resistance [25].

It is widely accepted that the increased controlled transcriptional expression of detoxing-related metabolic genes leads to an increase in the insect enzyme levels. For example, Bao et al. [26] detected the expression levels of two P450 cytopigments (*CYP6ER1* and *CYP6AY1*) related to imidacloprid resistance in several laboratory and field populations of brown planthopper and found that the P450 gene *CYP6ER1* in all the field populations was significantly overexpressed 7- to 24-fold, as compared to the laboratory susceptible strain. The study of Yang et al. [27] also found that the *CYP3, CYP4,* and *CYP6* families of *S. furcifera* were more closely associated with detoxification processes, and finally, resulted in a tolerance against imidacloprid. Although it is well-known that the dysregulation of the P450 genes played a key role in the process of insecticidal resistance, its molecular mechanism is still uncertain and needs to further explain its functions through some molecular technologies, such as RNA interference (RNAi). Liao et al. [28] specifically detected a significant overexpression of the P450 gene *CYP6ER1* in *N. lugens* (36.87-fold change) and then subsequently significantly increased the sensitivity of the drug-resistant populations towards sulfoxaflor through RNAi. Xu et al. [29] reported that multiple P450 genes could be associated with the resistance to chlorpyrifos and imidacloprid via feeding dsRNA in *N. lugens*. The study by Mao et al. [30] showed that the twelve P450 genes were disorders in the resistant strain of *N. lugens*, but when this resistance strain was injected with *dsCYP6ER1*, it returned to a susceptibility against nitenpyram.

In this study, two highly expressed P450 genes, *CYP408A3* and *CYP6CS3*, were selected from the XY17-G6 population (resistance ratio of 44.25-fold), which was obtained from a field resistant population and successively screened for six generations with the LD_50_ dose of chlorpyrifos. In addition, the full lengths of the two genes were cloned, and we conducted two gene interferences on XY17-6G through RNAi technology and then determined the susceptibility towards chlorpyrifos, and the activity of select P450 enzymes also was determined. Our preliminary results provided a preliminary insight into the expression changes of some P450 genes in WBPH during chlorpyrifos resistance and their relationship with regulation. The findings of this study are expected to be helpful in awareness of the WBPH resistance against chlorpyrifos, especially for the mechanism of resistance in the wild population after the mixed use of various pesticides.

## 2. Materials and Methods

### 2.1. Insects

The susceptible strains of *S. furcifera* (Lab-HN strain) (LC_50_: 3.262 μg/mL) were provided by Hunan Agricultural University. They were normally raised and domesticated in the laboratory for 15 years and have not been exposed to any pesticides and were uninterrupted when reared on rice seedlings (TN1) in our laboratory at a temperature of 26 ± 1 °C, relative humidity of 85% ± 10%, and photoperiod of 14 L:10 D. The field population (XY17) was collected in 2017 from Xuyong County, Sichuan Province, China.

The resistant population was collected from the field and screened with a chlorpyrifos LD_50_ dose (144.345 μg/mL) for 6 successive generations and proceeded as the resistant-chlorpyrifos population with 44.25-fold resistance.

### 2.2. Insecticides and Chemicals

Chlorpyrifos, a 98% technical product, was provided by Hubei Sharonda Co., Ltd. (Jingzhou, China). Triton X-100 was purchased from Chengdu Haobo Technology Co., Ltd. (Chengdu, China). DTT, PMSF, NADPH, and bovine serum albumin (BSA) were purchased from Beijing Solarbio Science and Technology Co., Ltd. (Beijing, China), as were EDTA-Na_2_ and Sodium Dodecyl Sulfate (SDS).

### 2.3. Bioassay

The indoor resistant population was screened by the rice seedling impregnation method, with little modification [16,31]. Third instar nymphs were used in the study. We soaked the rice seedlings and let them dry and wrapped the roots with cotton and placed them in a plastic cup. Fifteen third instar nymphs were released into each plastic cup that were reared on seedlings and cultured soilless in a rearing box incubator at 26 ± 1 °C, 85 ± 10% R.H., 14 L:10 D). The LC_50_ dose of the third instar nymph to chlorpyrifos was measured and calculated after 72 h of being treated. Finally, the LC_50_ dose was used to screen the strain, and the number of rice materials and nymphs were increased (30 rice seedlings, as many nymphs as pests).

### 2.4. Quantitative PCR (RT-qPCR) for Insecticide Resistant-Related P450 Genes

Total RNA for RT-qPCR of the Lab-HN strain and XY17 fifth- (G5) and sixth (G6)-generation populations were extracted by the traditional method by Chen et al. [31]. The CDS sequences of ten insecticide resistant-related P450 genes (*CYP6CS3*, *CYP6AX3, CYP408A3*, *CYP4CE3*, *CYP417A4*, *CYP4DD1*, *CYP18A1*, *CYP408A3*, *CYP6ER4,* and *CYP418A2*) were obtained from the research of Yang et al. [27], which referred to the transcriptome of *S. furcifera* by Wang et al. [32]; one reference gene (*RPL9*) [33] from the Lab-HN strain and the XY17 population each were amplified through RT-qPCR, with eleven pairs of corresponding primers (Appendix A). The mRNA levels of the P450 genes were measured by the methods of Wang et al. [34].

### 2.5. Cloning of CYP Genes Sequences

Full-length clones of P450 genes from resistant populations were adopted by the Kod-201 high-fidelity enzyme (200U Shanghai Toyang Textile Co., Ltd. (Japan)). The specific primers of two resistance-related P450 genes, *CYP408A3* and *CYP6CS3* [27], were designed based on the gene sequences from GenBank with Primer 3 online (Appendix A). After the P450 gene was cloned, its concentration and bands were detected by gel electrophoresis, and the gel was recovered.

### 2.6. Analysis of CYP Genes DNA and Protein Sequences

The prediction tool I-TASSER online (http://zhanglab.ccmb.med.umich.edu/I-TASSER, accessed on 10 April 2021) was used to predict the protein structure. PyMol 2.3.4 software was used for mapping and analysis. A phylogenetic tree was constructed with MEGA 7.0 software (MEGA, Tempe, AZ, USA) based on a multiple alignment of the amino acid sequences performed by Clustal X 2.0 [35] program software and by adopting the neighbor-joining algorithm with the bootstrap values determined by 1000 replicates.

### 2.7. Preparation of dsRNA

Before microinjection, the template was amplified with T7-specific primers to prepare the template (Appendix A). PCR products were analyzed on 1% agarose gel. Then, the PCR products were cloned and sequenced to confirm their identities and purified with VAHTS^®®^ RNA Clean Beads (Vazyme Biotech Co., Ltd. Nanjing, China). The dsRNA was synthesized by using a MEGA script^®®^ RNAi Kit (Thermo Fisher, Shanghai, China) according to the instruction manual and quantified by using an ultra-micro spectrophotometer (BIO-DL) at a wavelength of 260 nm. The calculated concentrations of the dsRNA are shown in Appendix A.

### 2.8. dsRNA Microinjection

To better ensure the interference efficiency, we explored the conditions of the microinjection, and the results showed that the nymph had the highest mortality rate, with forty percent for the third instar, and a minimum mortality rate, with twenty percent for the fourth instar nymph of white-back planthopper that was injected with a dose of 40 nL.

To ensure the efficiency of RNA interference, we used a UMP3/Nanoliter2010 micro-injection device (World Precision Instruments, Sarasota, FL, USA) to insert dsRNA synthesized in vitro into the fourth instar nymph, according to the method of Zhang et al. [36]. In addition, the conditions for appropriate microinjection parameters have been explored and determined that the amount of 40 nL from the total of 150 ng resulted in a minimum mortality rate of 20% and an appropriate interference efficiency of 80%.

The fourth instar nymphs of the XY17 population were anaesthetized by CO_2_ for 30 s and placed on agarose plates, and then, a pointed brush was used to place WBPH in the grooves, prepared in advance. The dsRNA solutions of 150 ng were injected into third instar nymphs after selecting the right injection sites with the proper strength. Then, they were reared in a greenhouse with a temperature of 26 °C (±1 °C), photoperiod of 14:10 h (L:D), and relative humidity of 85% (±10%). After rearing for 2 h, each of the BPHs was checked, whether it survived or not, and the rice seedlings were renewed after every three or four days. In this way, four treatments, including the *dsCYP408A3*, *dsCYP6CS3*, mixture of *dsCYP48A3* and *dsCYP6CS3*, and *dsGFP* diet (the blank control) were established with three replications. To assess the expression levels associated with each treatment, the whole bodies of the post-injection nymphs were collected at 24 h, 48 h, and 72 h for total RNA extraction and to calculate the reduction of the transcription levels of the P450 genes by using the RT-qPCR method, as above.

### 2.9. Assessment of Nymph Mortality and Cytochrome P450 Activity

To evaluate the RNAi efficacy of the *CYP408A3* and *CYP6CS3* genes in the susceptibility of the WBPH to chlorpyrifos, the injected nymphs (after 48 h of injection) were fed over the rice seedlings, treated with an LC_50_ dose of chlorpyrifos, and the mortality was recorded at 72 h post-treatment. Approximately, 30 injected nymphs were tested in each of the three replicates and the *dsGFP* as a control.

In order to evaluate whether two P450 genes (*CYP408A3* and *CYP6CS3*) mediate the normal survival of white-back planthopper (WBPH), we used indoor sensitive strains as verification materials, a microinjection of dsRNA synthesized by two genes in vitro into the fourth innermost nymphs, and then transferred them to fresh rice seedlings for feeding. The survival rate within 72 h after injection was observed and statistically analyzed. The experiment was set up in three replicates with 20 nymphs injected into each treatment.

To verify the RNAi effect on the two P450 detoxification genes, P450 activity was determined in the 4th instar nymphs from the Lab-HN strain and the XY17-G6 population. The total P450 activity was detected by Rose et al. [37] and Wang et al. [38]. About 30 nymphs were selected and injected with PBS buffer for tissue grinding. After, the fish were centrifuged at 4 °C at 15,000× *g* for 15 min using a 5417R centrifuge (Eppendorf, Hamburg, Germany). Finally, the supernatant was absorbed as the total crude enzyme solution and stored at a low temperature for testing. Then, 100 μL of 4-nitroanisole (2 × 10^−3^ mol/L) and 90 μL of crude enzyme liquid were mixed in a cell culture plate, followed by incubation for 3 min at 30 °C in a water bath. Next, 10 μL of NADPH (9.6 × 10^−3^ mol/L) was added into the system for a reaction. The changes in the OD value at the excitation wavelength of 380 nm and emission wavelength of 460 nm were recorded at the intervals of 45 s for 15 min (Spectramax i3x, Meigu Molecular Instruments (Shanghai) Co., Ltd. Shanghai, China). Finally, the specific activity of P450s was calculated using the standard curve of 7-hydroxycoumarin. The total protein content of the enzyme solution was determined by the Bradford [39] method while using bovine albumin as a standard.

### 2.10. Data Analysis

A Probit analysis was conducted by using the statistical analysis system (POLO 2.0) to calculate the slope, LC_50_, 95% CI, and χ^2^ values of chlorpyrifos to the susceptible strain and XY17 population for determining their resistance levels [34,40]. The relative expression of the P450 genes was calculated based on the 2^−ΔΔCT^ method [41]. The relative expression levels of the resistant-related P450 genes and mortality ratio were compared by using an analysis of variance (ANOVA), followed by a Student’s *t*-test for multiple comparisons (*p* < 0.05, 0.01), conducted by using SPSS (Microsoft 10.0)version 20.0 software.

## 3. Results

### 3.1. Expression Profiles of P450 mRNAs in Lab-HN and XY17 Populations of Sogatella furcifera

In insects, P450s play an important role in the metabolism of endogenous substances (such as juvenile hormones and ecdysis hormones) and exogenous substances (such as chemical pesticides). To analyze the molecular function of ten P450 genes in the development of resistance to chlorpyrifos, RT-qPCR was used to detect their expression levels in the resistant populations (G5 and G6). The results showed that the expression levels of the *CYP408A3* and *CYP417A4* genes were significantly enhanced in the XY17-G5 population 27.85-fold (*p* < 0.001) and 12.10-fold (*p* < 0.01), respectively, as compared to the Lab-HN strain. The relative expression levels of *CYP6FJ3* and *CYP6CS3* were 5.9- to 10.0-fold and were significant with the other genes (*p* < 0.05). For the other genes (*CYP6ER4* and *CYP4CE3*), the expression levels were slightly below the above genes (*p* < 0.05) and indicated 3.08-fold and 3.50-fold, respectively, while the P450 genes of *CYP18A1, CYP6AX3*, and *CYP4DD1* showed the minimum distribution in their expression levels, ranging from 1.34- to 2.38-fold (Figure 1A). For the gene *CYP6CS3*, its expression level was the highest in the third instar nymph population of XY17-G6, which was 26.13-fold, and the *CYP408A3* also had a high expression level (8.32-fold (*p* < 0.01)). The genes of *CYP4CE3* and *CYP6ER4* were still elevated at 3.64- and 4.01-fold, respectively, which was significantly higher than the other three genes: *CYP417A4*, *CYP18A1*, and *CYP4DD1*, ranging from 1.02- to 1.16-fold (Figure 1B). Therefore, *CYP408A3*, *CYP6CS3*, and *CYP6ER4* were significantly overexpressed in the XY17 G5 and G6-resistant populations. It can be speculated that the above genes may be related to the development of *S. furcifera* resistance to chlorpyrifos. Finally, the two genes of *CYP408A3* and *CYP6CS3* with the highest upregulation observed of the 10 genes that were screened were selected as the target genes for functional verification.

### 3.2. Characterization of Full-Length Amino Acid Sequences of Two Cloned P450 Genes

With multiple protein sequences alignments, there are four specific conserved motifs in the two cytochrome P450 genes (Figure 2). The heme–iron ligand signatures of the cytochrome P450 cysteine FxxGxRxCxG (*CYP408A3*: 448–457 and *CYP6CS3*: 447–456) were demonstrated in the lamina of the two cytochrome P450 proteins.

To further verify the diversity of the cytochrome P450 genes of *S. furcifera* during the resistance formation process, we compared the existing P450 sequence in the NCBI library with the evolutionary tree, as shown in Figure 3. The phylogenetic tree showed that *CYP6CS3* and *CYP408A3* were most closely related with *CYP6CS2v1* from *L. striatellus* and *CYP408A1v2* from *N. lugens,* respectively.

### 3.3. Interaction of the Tertiary Structure of CYP6CS3 and CYP408A3 with Chlorpyrifos

The optimal three-dimensional structure of the cytochrome P450 gene was predicted using I-TASSER online simulation (Figure 4A,B). The precheck results showed that 98.3%~98.9% of the amino acid residues in the three-dimensional (3D) structure of the cytochrome P450 genes were in the reasonable area of the Ramachandran point diagram, which theoretically indicated that the 3D structure of the two cytochrome P450 genes was reliable. After making a prediction about the optimization of the three-dimensional structure, we carried out the docking of the chlorpyrifos agent molecules with the model. The *CYP6CS3* and *CYP408A3* domain consists of several active amino acid residues and is located near the entrance of the active sac and at the heme-binding region. The total core value between *CYP6CS3* and chlorpyrifos was 6.3297 (crash of −1.4145). The total core value between *CYP408A3* and chlorpyrifos was 6.2386 (crash of −1.5198). In addition, the active structural cavity of the *CYP6CS3* protein of the P450 gene was significantly larger than that of the *CYP408A3* protein, and the molecular inclusion of chlorpyrifos was stronger, which provided more suitable conditions for the metabolic interpretation of chlorpyrifos.

### 3.4. Functional Analysis of CYP6CS3 and CYP408A3 via RNAi

To elucidate the functions of *CYP6CS3* and *CYP408A3*, two P450 genes in *S. furcifera* in mediating the development of chlorpyrifos resistance, the relative expression levels were detected via RT-qPCR in the fourth instar nymphs of the XY17-G6 population at 24 h, 48 h, and 72h after the dsRNA injection. The results showed, after the injection, that the expression of *CYP6CS3* and *CYP408A3* decreased within 48 h, as compared to the *dsGFP* injection, in which the most obvious results, which was 93.45% for injected *CYP6CS3* and 80.34% for *CYP408A3* at 72 h (Figure 5), respectively (*p* < 0.01), were observed. In addition, the mRNA expression level of the injection containing the mixture of the two genes was also significantly decreased, which was stable at about 94% (*p* < 0.01).

The larvae that survived well in each treatment were selected at 48 h post-dsRNA injection, and the P450s enzyme activities were determined. The results showed that the P450 activity level of the XY17-G6 population (3.42 nmol∙min^−1^∙mg pro^−1^) and XY17-dsGFP treatment (2.96 nmol∙min^−1^∙mg pro^−1^) were consistent and significantly higher than that of the Lab-HN strain (2.42 nmol∙min^−1^∙mg pro^−1^) (*p* > 0.05). In target gene interference processing, the P450 activity in the larvae of the *dsCYP6CS3* treatment (2.08 nmol∙min^−1^∙mg pro^−1^) showed a significant reduction. For the *dsCYP408A3* (2.24 nmol∙min^−1^∙mg pro^−1^) and mixture of *dsCYP6CS3* and *dsCYP408A3* (2.29 nmol∙min^−1^∙mg pro^−1^) injection treatments, they also showed a significant reduction compared with the other treatments, but it was slightly higher than that of the *dsCYP6CS3* treatment (Table 1).

To assess the efficiency of RNAi, the survived fourth instar nymphs after 48 h of injection, containing the dsRNA of *CYP6CS3* and *CYP408A3*, were dealt with LC_50_ of chlorpyrifos. The results showed that the mortality rate was increased significantly in the RNA interference treatment group compared with the treatments of injecting *dsGFP* (51.67%) and G6-Blank control (48.9%) generation (*p* < 0.01). Among the treatments of RNA interference, the mixture of *ds**CYP6CS3* and *ds**CYP408A3* and sole *dsCYP6CS3* had the highest mortality (93.33% and 91.67%, respectively) at 72 h after injection (*p* < 0.05), followed by the treatment of sole *dsCYP408A3* reaching 69.33% (Figure 6).

In order to evaluate whether the selected genes were necessary for the survival of the test insect, we injected dsRNA into the indoor sensitive population to observe the mortality rate of the test insect. The results of the functional necessity validation of the two P450 genes in the HN-Lab showed that the nymphs in the *CYP408A3* and *CYP6CS3* injection groups had good survival compared with the blank control group (95%). The survival rate of the nymphs in the *dsGFP* and *dsCYP408A3* injection group was 93.3%. The survival rate of the *dsCYP6CS3* injection group was 91.6% (Figure 6).

All results suggested that the RNAi-mediated silencing of *CYP6CS3* increased the susceptibility of the XY17-G6 population to chlorpyrifos.

## 4. Discussion

Chlorpyrifos, as a commonly utilized effective insecticide, has been used in China for more than 15 years [42]. It can be used for effective control of rice pests because of its high capability of internal absorption and acts as an acetylcholinesterase inhibitor in insects [13]. Like all traditional pesticides, under the conditions of excessive or irregular use, chlorpyrifos resistance is still produced in various rice pests, including *S. furcifera* [43,44]. Under natural conditions, there are many possible reasons for the development of resistance in *S. furcifera* [45]. However, the increase of the metabolic activities of the different detoxification enzymes is still an important factor, promoting the development of insecticidal resistance. The reports of Xu et al. showed that three P450 genes were significantly overexpressed (6.87–12.14-fold) in the chlorpyrifos resistant population of *L. striatellus* [29]. In addition, it was proved by a RNAi experiment that the resistance in the indoor resistant population of *Nilaparvata lugens* against clothianidin was caused by the overexpression of the P450 gene *CYP6ER1* [46]. According to the research of our lab, P450 *CYP6ER4* was screened by RT-qPCR for its field resistance to the chlorpyrifos *S. furcifera* population, and RNA interference was used to verify that the gene was mainly involved in the generation of resistance to chlorpyrifos in the population [47]. In the present research, we detected the upregulation of six CYP genes among ten tested in the XY17-resistant population, which proved that the P450 detoxification enzyme could mediate the metabolic resistance of the white-backed planthopper to chlorpyrifos.

Currently, it has been accepted by pesticide toxicity researchers that the development of pesticide resistance is caused by the controlled overexpression of resistance-related detoxification genes such as P450 under continuous pesticide stress [48,49,50,51]. However, there are a large number of P450 genes that may be combined by several genes when they perform various functions in insects [45]. Zhang et al. [24] found that the twelve P450 genes were observed for upregulation in the imidacloprid-resistant *N. lugens* population. Four P450 genes, *CYP6AY1*, *CYP6ER1*, *CYP4CE1*, and *CYP6CW1*, played important roles in imidacloprid resistance after RNAi and in vitro recombination of the corresponding proteins. One more research showed that nine P450 genes were upregulated and three P450 genes were downregulated under the LD_85_ dose of Imidacloprid and Cycloxaprid [27]. Ali et al. [15] also found that sixteen P450s and one GST gene were significantly overexpressed in the dinotefuran-resistant strain (22-fold) two-fold higher than the susceptible strain. In our study, while detecting the spatial and temporal expression of ten resistant P450 genes, among which four P450 genes in the CYP6 Clade (*CYP6ER4, CYP6FJ3, CYP6AX3*, and *CYP6CS3*); four P450 genes in the CYP4 Clade (*CYP4CE3*, *CYP417A2*, *CYP4DD1*, and *CYP408A3*); and one in the CYP2 Clade (*CYP18A1*) were upregulated in the XY17-resistant population, especially for *CYP417A2, CYP408A3*, *CYP6ER4*, and *CYP6CS3.* These results are consistent with the mechanism proposed by Okey that there are different expression levels of multiple P450 genes during the development of resistance [52]. As a mature sequencing technology, RNA-Seq provides real-time and stable technical support for researchers in the process of mining drug resistance-related genes [53]. However, for transcriptome sequencing, materials with a stable genetic background are required. Our experimental design was defective due to the different backgrounds of the chlorpyrifos-resistant population in the field and sensitive population in the room. In the future, we will use a two-way screening method to screen sensitive and resistant populations, so as to better use transcriptome sequencing to reveal the expression of resistance genes. RNAi is an effective technique with high specificity in the last decade and is being widely used to investigate the gene functions in insects [54,55]. We investigated the contribution of multiple P450 genes that were expressed significantly in the formation of pesticide resistance and their interactions by the RNAi technique. When the expression levels of *CYP6CS3* and *CYP408A3* were suppressed by applying the injection of *dsCYP6CS3*, *dsCYP408A3* and *dsCYP6CS3* and *dsCYP408A3*, the mortality of the larvae treated with the LC_50_ of chlorpyrifos increased significantly, especially for the injection having a mixture of two genes, which showed higher mortality rates, and this phenomenon was consistent with the research results of Lu et al. [45]. Moreover, the total P450 activity of every treatment showed a significant reduction as compared with the control, but for the mixed injection treatment, it was slightly lower than the single gene interference effect, probably because we used the half-concentration of a single gene from the total injection when we injected the dsRNA mixture. However, there was a slight downregulation of the P450 enzyme activity in the *dsGFP* injection group, which may be due to the fact that the nymphs themselves needed a certain recovery time after the microinjection, so the P450 enzyme activity in vivo was in a relatively unstable state. It was worth considering about these two P450 genes may not be necessary for the survival of *S. furcifera*, and due to that, the treated nymphs almost never died after we injected dsRNA alone into the HN-Lab population without exposure to chlorpyrifos. These results indicated that the development of resistance in the XY17 population is mediated by the two or more pairs of multiple P450 genes, and there is a cascade regulation and synergistic effect between multiple genes that still need to be further studied to clarify and explore the regulatory mechanisms.

The P450s are multi-enzymatic complexes and have a long history; the diversity of the protein structure determines the diversity of the function. The structural differences of the insect cytochrome P450 protein can reflect the evolutionary process of the P450 gene. At present, as an inherent functional domain of the P450 gene, the heme-binding domain FxxGxRxCxG is the main target to explore the structure and function of the P450 gene [56]. Our research showed that the conserved region was located near the typical active lumen and the molecular modeling display and that structure of chlorpyrifos was surrounded by the active site of *CYP6CS3*. The protein has a predicted oval active site structure, large volume, and large substrate channel, enabling chlorpyrifos to adapt to active site cavities. The spacious cavity of the P450 enzyme allows larger molecules to access the oxygen bound to the heme in the reaction center and brings the substrate closer to the binding site [57]. Our results are close to the research of Rupasinghe [58], in which a comparison of the homology models for *CYP688* and *CYP321A1* described their substrate-binding cavities, which were predicted to be more spacious in these two enzymes that metabolize a wider range of compounds. Therefore, we hypothesized that *CYP6CS3* can present a greater metabolic ability than *CYP408A3* [59,60]. This result also confirmed the phenomenon that *dsCYP6CS3* had a high interference efficiency and a high degree of P450 enzyme activity decline mediated by *CYP6CS3*. Moreover, the P450 gene is ubiquitous in all species. Although its gene functions are diverse, the P450 gene forms some conserved regions in the process of species adaptation and evolution to maintain a physiological balance for insects when they respond to external stress. In our research, the phylogenetic tree comparison showed that the two selected resistance-related P450 genes were close to their homologous resistance genes. The results showed that gene *CYP6CS3* was most closely related with *CYP6CS2v1*, which is from the deltamethrin-resistant *L. striatellus* strain [61], and *CYP408A3* was most closely related with *CYP408A1v2* from the *N. lugens* strain [62], respectively. From the function of the related subfamily genes, it can be analyzed that the two P450 genes selected by us have a high possibility of benig involved in the formation of metabolic resistance in resistant populations.

Based on the results of our study, we speculated that the P450 genes *CYP408A3* and *CYP6CS3*, especially *CYP6CS3*, play an important role in *S. furcifera* against chlorpyrifos resistance. Additionally, further investigation is needed to determine which P450 is the key gene involved in the development of resistance in white-backed planthoppers. Our results lay the foundation for further exploring the functions of *CYP408A3* and *CYP6CS3* in resistance formation through prokaryotic expression, such as in vitro prokaryotic expression and eukaryotic expression, and the function of the two genes can also be verified by further gene knockouts.

## 5. Conclusions

In this study, we detected the P450 gene expression in a field chlorpyrifos-resistant population screened for six generations with a LD_50_ dose (XY-17), and two selected P450 genes were verified by RNAi. Our results showed that the P450 gene *CYP6CS3* may play an important role in the development of resistance to chlorpyrifos in this population. This work could help to deepen the understanding of chlorpyrifos resistance research in white-back planthoppers (WBPH) and could also inform decision-making for the development and management of chlorpyrifos resistance in wild WBPH.

## Figures and Tables

**Figure 1 biology-10-00795-f001:**
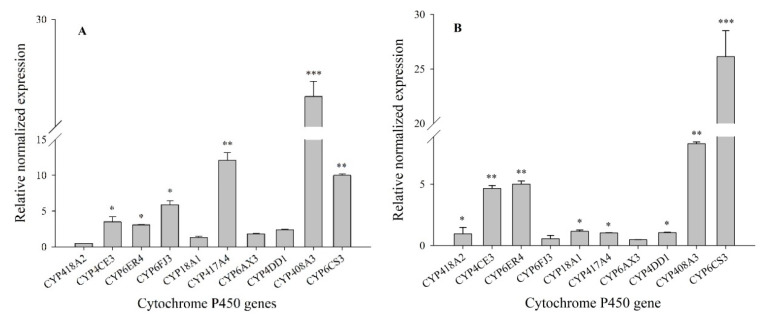
Relative expression qualities of select insecticide resistant-related P450 genes in the 3rd instar nymph resistant populations selected by a LC_50_ dose of chlorpyrifos for 5 generations (**A**) and 6 generations (**B**) compared to the Lab-HN strain. The relative normalized expression was presented as the mean of three replications ± SE. The F_9, 20_ values of the relative expression qualities of insecticide resistant-related P450 genes in the 3rd instar nymph of the XY17-G5 (**A**) and XY17-G6 (**B**) field populations were 378.54 and 100.745, and the *p*-values were = 0.0001 < 0.01. *, **, and *** showed significance at the 0.05, 0.01, and 0.001 levels with a Student’s *t*-test, respectively.

**Figure 2 biology-10-00795-f002:**
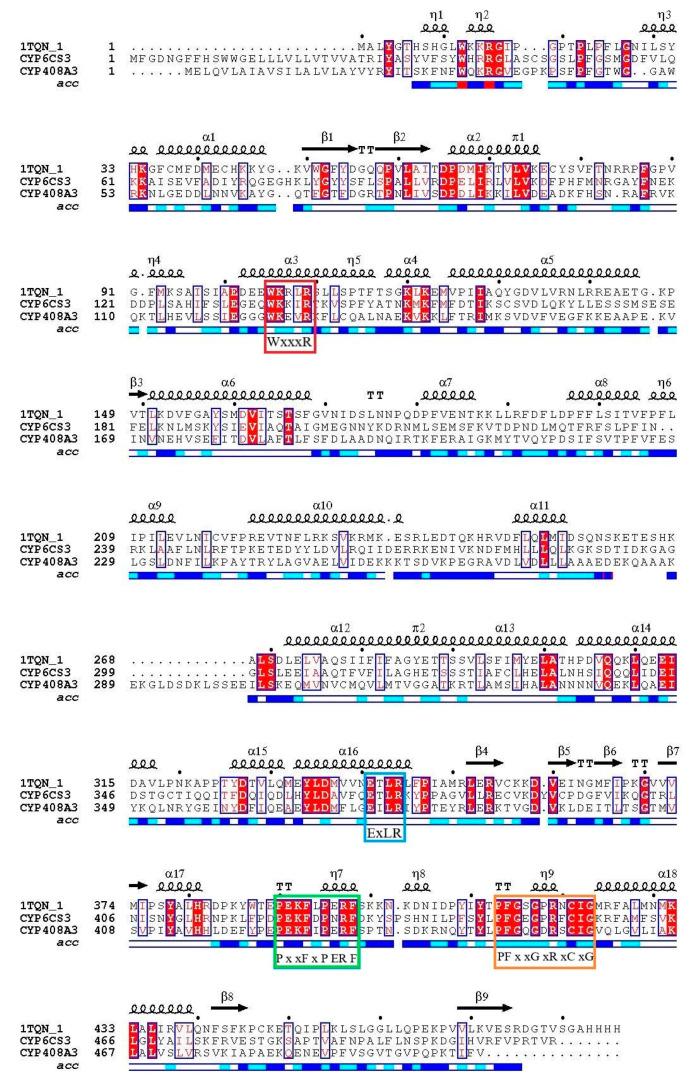
Graph of sequence alignment in three cytochrome P450 genes. The 1TQN-1 sequence is a reference sequence for predicting the function of two amino acid sequences of the P450 gene. The red box located in helix 4 is a conserved sequence, and the amino acid sequence is WxxxR; the blue box located in helix 15 is the conserved sequence, with the amino acid sequence ExLR; the green box of the amino acid sequence is PxxFxPE/DRF; the orange box located in random coil 13 (13, heme binding) is a conserved sequence; and the amino acid sequence is FxxGxxxCxG.

**Figure 3 biology-10-00795-f003:**
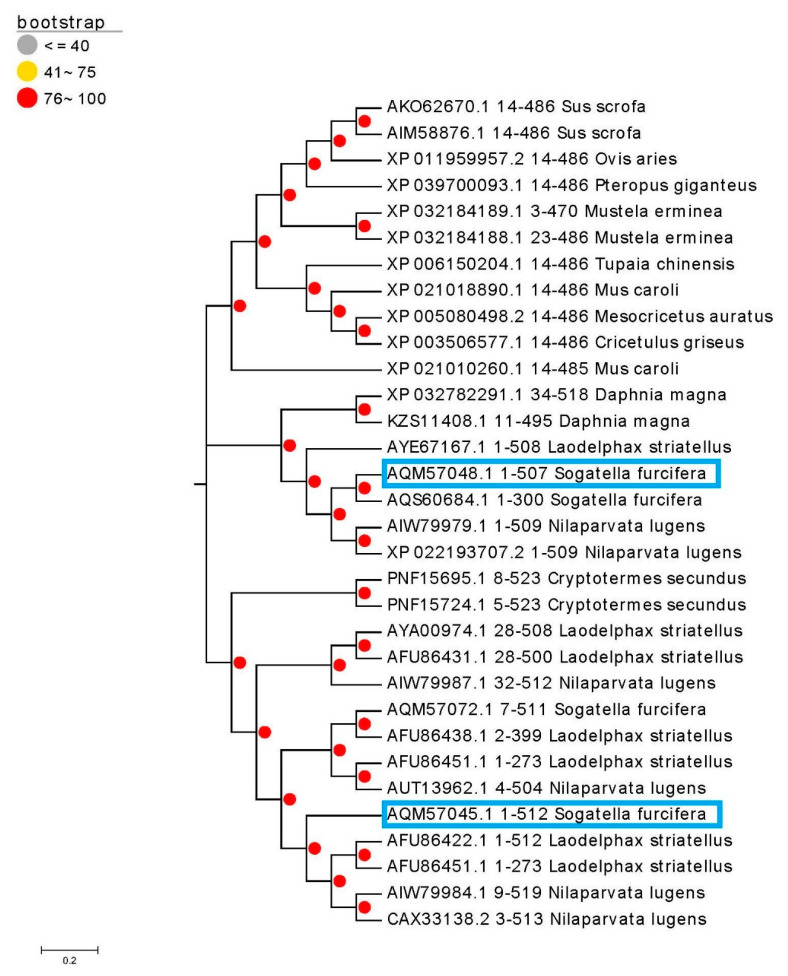
Phylogenetic relationship among *CYP6CS3* and *CYP408A3*, other insect P450s. Two selected and validated P450 genes have been labeled with blue boxes (The AOM57048.1 represents the P450 gene CYP408A3 and the AOM57045.1 represents the CYP6CS3). The GenBank accession numbers are shown before the P450 names. The phylogenetic tree was inferred using Neighbor-Joining. The scale bar indicates 0.1 amino acid substitutions per site. The percentage of the replicate trees in which the associated taxa clustered together in the bootstrap test (1000 replicates) is shown next to the branches. The phylogenetic analyses were conducted in MEGA 7.0.

**Figure 4 biology-10-00795-f004:**
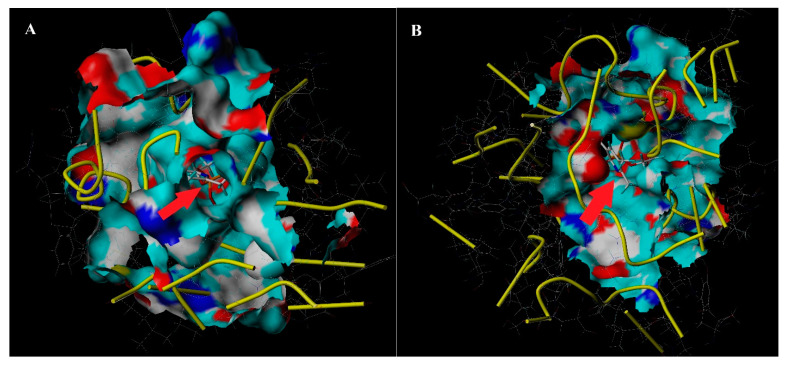
The docking structure with chlorpyrifos of *CYP6CS3* and *CYP408A3*. (**A**) *CYP6CS3* domain and chlorpyrifos. (**B**) *CYP408A3* domain and chlorpyrifos. The molecular cartoons were drawn with PyMol 2.3.4 software. The molecular model of chlorpyrifos was embedded in the active cavity of the P450 protein molecule, and the size of the opening of the active cavity affected the metabolic activity of the corresponding P450 protein (the active cavity is indicated by the red arrow).

**Figure 5 biology-10-00795-f005:**
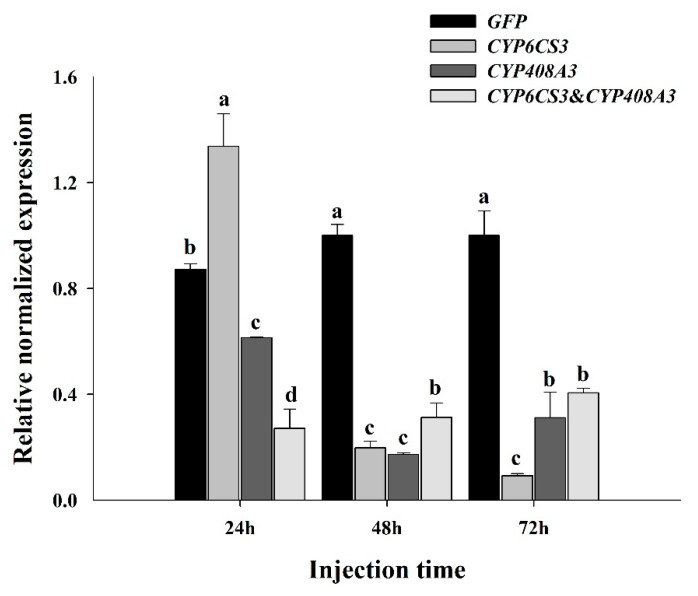
Functional analysis of *CYP6CS3* and *CYP408A3* by RNAi. The relative expression of *CYP6CS3* and *CYP408A3* in the fourth instar nymphs injected with *dsCYP6CS3*, *dsCYP408A3*, or the mixture of *dsCYP6CS3* and *dsCYP408A3* in each period compared with *dsGFP*. Significant differences are indicated by different letters, for example, a, b, c, and d (*p* < 0.05, with a Student’s *t*-test).

**Figure 6 biology-10-00795-f006:**
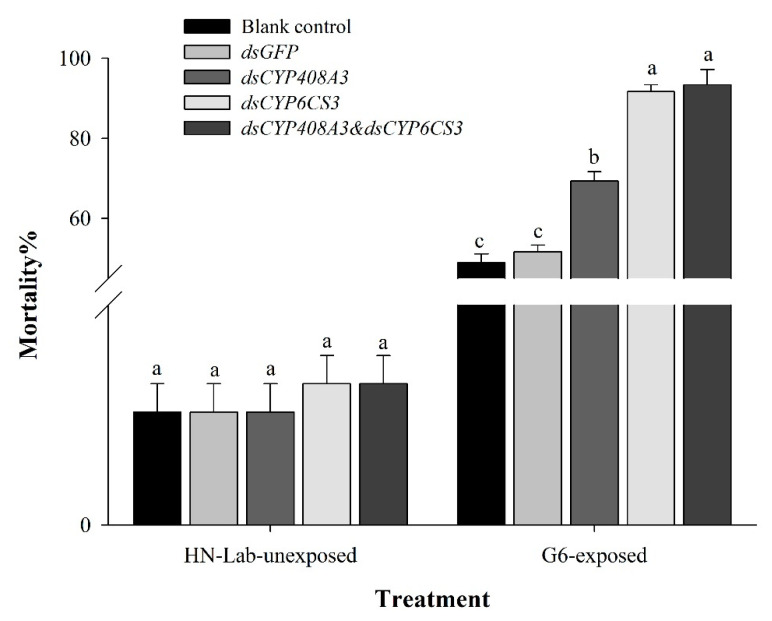
Mortality at 72 h of the dsRNA-injected fifth instar nymphs after being unexposed to the chlorpyrifos in the HN-Lab and exposed to the chlorpyrifos in the XY17-G6 population. Significant differences are indicated by different letters, for example, a, b, and c (*p* < 0.05, with a Student’s *t*-test).

**Table 1 biology-10-00795-t001:** P450 activity of the Lab-HN strain and XY17 population 48 h after RNAi.

Treatment	P450 Activity		SR ^a^
	nmol·(min·mg·pro)^−1^		
Lab-HN	2.42 ± 0.007	c	1.0
XY17-G6	3.42 ± 0.02	a	1.41
XY17-G6-*dsCYP6CS3*	2.08 ± 0.04	e	0.86
XY17-G6-*dsCYP408A3*	2.24 ± 0.19	d	0.93
XY17-G6- *dsCYP6CS3* & *dsCYP408A3*	2.29 ± 0.07	cd	0.95
XY17-G6-*dsGFP*	2.96 ± 0.02	b	1.22
	F_4, 10_ = 184.468, *p <* 0.001		

^a^ SR (synergistic ratio) = the enzyme activity of XY17-G6 or XY17-G6 treated with RNAi/the enzyme activities of Lab-HN.

## Data Availability

The authors confirm that the data supporting the findings of this study are available in publicly accessible repositories within the bibliography.

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
