# Peer review of "Cloning and Functional Verification of CYP408A3 and CYP6CS3 Related to Chlorpyrifos Resistance in the Sogatella furcifera (Horváth) (Hemiptera: Delphacidae)"

_biology, 2021, doi:10.3390/biology10080795_

Round 1

Reviewer 1 Report

In the present revised manuscript by Ruan et al., the authors have satisfactorily addressed all the concerns previously raised. However, a few minor points should be corrected before acceptance.

Minor concerns:

  1. Lines 20-21, “After picked out the two highest expressions of genes, the function of these two genes in the process of pesticide resistant was verified through RNA interference”, this sentence would be clearer if it were worded as follows: “After selecting the two genes with the highest expression, the function of these two genes in the pesticide resistance process was verified by RNA interference”.
  2. Line 101, the words “we used” should be deleted.
  3. Line 144, please add a coma before “which”.
  4. Lines 177-182, please do not italicize.

Author Response

Dear Editor and Reviewers,

   Thank you for your comments on our manuscript. We have revised our manuscript according to the review's suggestion. We have carefully reviewed all comments and suggestions and answered those questions point-by-point and expected the revised manuscript was acceptable for publication.

Comment #1: Lines 20-21, “After picked out the two highest expressions of genes, the function of these two genes in the process of pesticide resistant was verified through RNA interference”, this sentence would be clearer if it were worded as follows: “After selecting the two genes with the highest expression, the function of these two genes in the pesticide resistance process was verified by RNA interference”.

Answer: Agree. We have exchanged the sentence to “After selecting the two genes with the highest expression, the function of these two genes in the pesticide resistance process was verified by RNA interference” in the new manuscript.

Comment #2: Line 101, the words “we used” should be deleted.

Answer: Agree. We have deleted the words “we used”.

Comment #3: Line 144, please add a coma before “which”.

Answer: We have added a coma before “which”.

Comment #4: Lines 177-182, please do not italicize.

Answer: We have fixed the font formatting.

We look forward to your positive response.

Yours sincerely,

Dr. Wang

Reviewer 2 Report

Thank you for addressing the issues that were brought forth.

Author Response

Dear Editor and Reviewers,

Thank you for your comments on our manuscript. We have revised our manuscript according to the review's suggestion. We have carefully reviewed all comments and suggestions and answered those questions point-by-point and expected the revised manuscript was acceptable for publication.

Reviewer 3 Report

Ok to be accepted.

Author Response

Dear Editor and Reviewers,

Thank you for your comments on our manuscript. We have revised our manuscript according to the review's suggestion. We have carefully reviewed all comments and suggestions and answered those questions point-by-point and expected the revised manuscript was acceptable for publication.

This manuscript is a resubmission of an earlier submission. The following is a list of the peer review reports and author responses from that submission.

Round 1

Reviewer 1 Report

The present manuscript by Ruan et al. is a revised version of a previous submission, which has been considerably shortened. The lack of line numbering and clear indication in the answer to reviewer's comments of the changes made in the revised manuscript greatly annoyed the reviewer and made his job much more difficult. It is so simple...

However, significant modifications have been done to the manuscript and the authors tried to answer all concerns. Unfortunately, they did not go far enough in this revision and several concerns remains unsatisfactorily addressed and many language errors remain. In addition, several of the arguments made in the response to the reviewer should be included in the manuscript.

In what follows, the reviewer will reuse the numbering of remarks used by the authors in their response.

Comment #1: The reviewer acknowledges that former Fig.5 has become Fig.2, which is more explicit for the reader. However, the lack of even a weak match between the sequences characterized in this work and the genome in the database is surprising and concerning. How can the genetic background be so different to explain this? Is it really the same species? This point should definitely be mentioned and explained in the discussion.

In addition, the authors continue to omit the reference to this genome (Wang et al. Gigascience 2017 6(1):1-9; doi: 10.1093/gigascience/giw004). Is there any reason for this? It is the third time I ask for it!

Comment #2: The authors do not answer to the concern. The control, i.e. unexposed organisms transfected by RNAi, is missing. The reviewer reiterates his request to perform additional RNAi experiments with the control laboratory strain (Lab-HN) to determine whether or not the observed increase in mortality (Fig. 6) may be due to CYP6CS3 knockdown independent of chlorpyrifos exposure.

Comment #3: The authors' response is somewhat disturbing about the relevance of their experimental protocol. At the very least, the explanations given by the authors should be mentioned in the article and clearly identified as a limitation in the interpretation of the results.

Comment #4: The reviewer can understand that the authors do not want to make DNA array or RNAseq experiments at this time. However, this is clearly unavoidable nowadays. The answer given by the authors should be present at the end of the discussion to open up perspectives; i.e. “In addition, the genetic background of the sensitive population used in the current experiment is different from that of the existing resistant population. In the future, we will continue to conduct two-way screening against the resistant population to obtain the resistant degenerated population and the highly resistant population, and the differentially expressed genes may be further screened through Microarray or RNASEQ transcriptomic analyses in the future to verify gene function.” Of course, this has to be slightly modified to be integrated in the discussion.

Comment #10: Although the original sentence has disappeared from the revised manuscript, the question remains unanswered. So, regarding CYP408A3 and CYP6CS3, were these two genes the ones with the highest expression or only two with high expression levels? This should definitely be clarified in the manuscript.

Comment #15: The authors do not address this concern or have misunderstood it. In the sentence “To further verify the diversity of resistance to P450 in S. furcifera, we compared the existing P450 sequence in the NCBI library with the evolutionary tree as shown in Figure 3.”, do the words “the diversity of resistance to P450 in S. furcifera” mean “the diversity of Cytochrome P450 genes”? If yes, please change. If not, please explain what is the diversity of resistance to P450.

Comment #20: The authors made the required change, but without paying attention: the word "term" after Neighbor Joining should be deleted (legend to Figure 3).

Moreover, this figure was of much better quality in the previous version of the manuscript and especially the previous one included the bootstrap value (as colored dots), which have disappeared here. Please reuse the old version of this phylogenetic tree.

Comment #21-23: The authors have modified the sentences but the English language need to be carefully edited.

Additional concern: The Supplementary Table 1 is actually two separate tables. Therefore, please revise the numbering of Supplementary Materials and make the corresponding update in the manuscript.

All others minor concerns (#5-9; 11-14; 16-19; 24-34) have been more or less satisfactorily addressed.

Reviewer 2 Report

Dear editor,
The authors made all the suggested modifications. The article is innovative and will undoubtedly bring important contributions to the management of pest insects.
Sincerely your,

Reviewer 3 Report

  1. Why were these 10 p450s specifically selected?
  2. Potential for insecticide/pesticide resistance is not necessarily inherent unless those that are used are misused or overused.
  3. There is no discussion/acknowledgement that up- and down-regulation are both significant factors related to p450 expression and the development of resistance. An organism can also down-regulate enzymes that are necessary to break down specific pesticides (i.e., another strategy).
  4. There are no values on the phylogenetic tree so the reader can't decipher if the finding are significant or relevant. If the branch points have a low percentage of accuracy then the findings are largely irrelevant.
  5. There is still a significant amount of English wording and spelling that needs to be corrected.

Reviewer 4 Report

Authors shown that the CYP408A3 and CYP6CS3 are involved in chlorpyrifos resistance, applying also quite elaborative method as RNAi (dsRNA).

Minor points

  • In the text formatting the line numbers and even page numbers are missing.
  • Yellow highlights in the title and in the text: have they any meaning or just typos?
  • The abbreviations are not clear starting from the Abstract (e.g. XY17-G5 meaning, Sus-Lab, etc.)

Major points

  • In Methods, page 12, “To verify the RNAi effect on the two P450 detoxification genes, P450 activity was determined …” Please, describe the activity assay more precisely due to its importance, not only the citations. Is it possible with this method to recognize the individual activity of each CYP408A3 and CYP6CS3? It looks like the assay recognized non-specifically the activity of ALL P450 cytochromes from the insect, right? For the activity assay the spectrophotometry spectra must be reported in this paper.
  • Why the mortality in controls (G6 Blanc control, Figure 6) is so high as 50%. Is it normal for Lab quality rearing?